# SPARSELY MULTIMODAL DATA FUSION

## ABSTRACT

Multimodal data fusion is essential for applications requiring the integration of diverse data sources, especially in the presence of incomplete or sparsely available modalities. This paper presents a comparative study of three multimodal embedding techniques, Modal Channel Attention (MCA), Zorro, and Everything at Once (EAO), to evaluate their performance on sparsely multimodal data. MCA introduces fusion embeddings for all combinations of input modalities and uses attention masking to create distinct attention channels, enabling flexible and efficient data fusion. Experiments on two datasets with four modalities each, CMU-MOSEI and TCGA, demonstrate that MCA outperforms Zorro across ranking, recall, regression, and classification tasks and outperforms EAO across regression and classification tasks. MCA achieves superior performance by maintaining robust uniformity across unimodal and fusion embeddings. While EAO performs best in ranking metrics due to its approach of forming fusion embeddings post-inference, it underperforms in downstream tasks requiring multimodal interactions. These results highlight the importance of contrasting all modality combinations in constructing embedding spaces and offers insights into the design of multimodal architectures for real-world applications with incomplete data.

## 1 INTRODUCTION

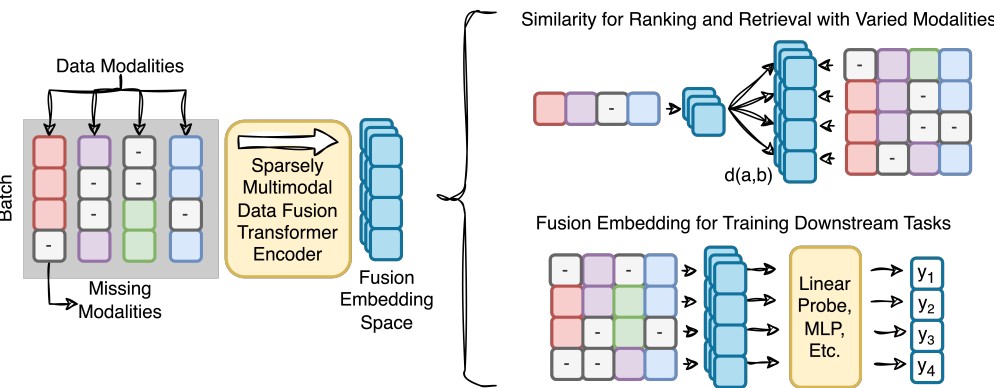

Figure 1: An overview of the main motivation and purpose of this study, where multimodal datasets (in this case, 4 modalities) that have samples with missing modalities can be encoded into a fused embedding space. The embeddings are used to perform both ranking and retrieval tasks, as well as for downstream regression and classification tasks.

Multimodal data is becoming the norm for deep learning applications.(Xu et al., 2023; Han et al., 2023; Liang et al., 2022a) Many models are trained on data with two aligned modalities(Alayrac et al., 2020; Fei et al., 2022; Huang et al., 2024; 2021; Hager et al., 2023), including incorporating images into large language models.(Alayrac et al., 2022; Rahman et al., 2020) Models with more than two aligned data modalities have also become well studied,(Mizrahi et al., 2024; Srivastava & Sharma, 2024; Akbari et al., 2021), and recent examples have explored learning from multiple unaligned or partially aligned data modalities(Yang et al., 2021; Tran et al., 2023; Wei et al., 2023; Nakada et al., 2023).

Most of these examples use a combination of two modalities of text, audio, image or video. However, applications can use data other than these traditional media formats. For example, multisensor fusion in home monitoring systems and robotics includes tabular sensor data and time series data from different types of sensors.(Tonkin et al., 2023). Bioinformatics and biomedical applications use data that consists of tabular, image, and sequence data. In these fields, each media format in the data can also be comprised of different modalities having the same data type but different data source, for example as in two tables of data from different types of experiments.(Cui et al., 2023; Lynch et al., 2022) In examples like these, datasets with 3 or more modalities are able to be constructed.

As the number of modalities used for training a model increases, samples with missing modalities are more likely to occur, which are called modal-incomplete samples in this study. Multimodal fusion models which cannot use modal-incomplete samples may not be well suited to be used for applications with datasets that have many modalities because the likelihood of missing modalities increases. When a significant fraction of samples are modal-incomplete, a dataset is here referred to as sparsely multimodal. This contribution explores the extent to which the amount of modal sparsity, defined in METHODS, affects multimodal fusion models based on contrastive representation learning.

Constrastive learning with multimodal data describes an important class of multimodal model.(Liang et al., 2022b; Shvetsova et al., 2022; Singh et al., 2022; Akbari et al., 2021; Noriy et al., 2023; Radford et al., 2021) In the well-known CLIP model, it generates joint embeddings for text and images which can be used for classification, regression or for conditioning of diffusion probablistic models. Embeddings generated from contrastive learning can be used for multimodal retrieval, such as in the Everthing At Once model (EAO)(Shvetsova et al., 2022) where video retrieval can be performed via text or audio via a fused embedding space. Recent studies have shown that using contrastive learning between unimodal representations and a learnable fusion representation can generate useful fusion embeddings. Originally presented as the Zorro model, this type of model uses block attention masking to effectively perform self-attention on unimodal representations and attention to a fused representation, all in a single attention block.(Shi et al., 2023; Recasens et al., 2023)

In this study, an attention masking and contrastive representation learning strategy called Modal Channel Attention (MCA) was devised which combines aspects of EAO and Zorro. MCA improves on retrieval metrics compared with Zorro and improves on downstream task performance when compared with both Zorro and EAO. This is demonstrated using two well-known datasets, each with four modalities. The effect of sparsely multimodal data on each model's performance is compared throughout.

## 2 COMPARISON OF RELATED WORK

In this section, existing approaches to train models on datasets with modal-incomplete samples are described. First, relevant models which do not use contrastive loss, then those which include contrastive learning as a component, followed by those using only contrastive loss. Then, justification for the use of a contrastive learning for the purposes of this study is given and a comparison of existing contrastive models to MCA.

Without contrastive loss, interleaved data can include incomplete modalities due to the treatment of samples as a sequence.(Alayrac et al., 2022) Similarly, masking of representations in a late stage fusion block can be used naturally with modal-incomplete data(Zhang et al., 2022; Tran et al., 2023). These model architectures require an autoregressive or a masked language objective in order to train on the interleaved data or predict missing data. We do not explore these classes of models here and instead constrain the approach to explore those using a contrastive learning objective.

FLAVA(Singh et al., 2022) uses multiple loss models for data which is missing modalities. For text, they use a masked language objective, while for image-text pairs, they use a contrastive loss. Their approach allows for unimodal and multimodal inference, but does not generate a multimodal fusion embedding space. Zhang et al. (2023) developed a model which can be trained with missing modality combinations by projecting unimodal encodings into a modality-aligned feature space. They then perform weight-shared dual attention prediction of two sets of outputs. The first output is trained with supervision to class labels, while the second prediction is trained with supervision to unimodal

predictions across epochs. This is shown to improve predictions for unseen modalities. LORRETA uses an objective of predicting a third modality given two other modalities. This approach requires bimodal pairs for each forward pass and can not directly embed higher order modality combinations.(Tran et al., 2023) A similar objective of predicting missing modalities was taken in Wei et al. (2023).

None of these aforementioned models generate a fusion embedding space, which is required for tasks based on embeddings like retrieval and linear probing for regression and classification. There is no clear extension to the aforementioned models where a fusion embedding is close to it's unimodal embeddings and to embeddings generated from modal-incomplete samples. Since the goal of this study is to explore fused embedding spaces for data with more than 2 modalities when the datasets are sparsely multimodal, we now turn to models designed for related purposes.

The Everything At Once (Shvetsova et al., 2022) model uses a transformer encoder that creates embeddings for one or two modalities at a time with contrastive loss. The encoder is applied multiple times per minibatch in order to formulate embeddings for each modality and each pair of modalities. These embeddings are applied in a combinatorial contrastive loss function, such that the loss is applied to each possible pair of generated embeddings. At inference time, the generated embeddings are averaged to create a fusion embedding. This method requires a number of forward passes that has unfavorable scaling with the number of modalities. Importantly, it does not attend jointly from all embeddings at once to form a unified fusion representation.

Zorro is a transformer encoder model which produces both unimodal embeddings and a fusion embedding which are then trained with contrastive loss.(Recasens et al., 2023) Zorro uses block attention masking to prevent attention between the internal representations of different modalities, but allows for unimodal self-attention and attention from unimodal representations to a fusion representation. A similar architecture was recently applied by Shi et al. in order to fuse 3 modalities for image segmentation in a biomedical application.(Shi et al., 2023) In this latter study, it was noted that the model architecture seemed to function well even with missing modalities, but it did not directly explore the performance of this type of modal fusion with sparsely multimodal data.

In publications presenting the Zorro and EAO models, three common data modalities (text, audio, and video) were used. However, the model principles can be applied to any type and number of modalities. An overview of how these models are related for 2 and 3 modality datasets is shown in Figure 2. When applied to 2 modality datasets, EAO, Zorro, and MCA are conceptually similar. All models separately contrast each of 2 generated unimodal embeddings with a fusion embedding. The most important difference between EAO and the other models in the case of 2 modalities is that the unimodal embedding is created in the same forward pass as the fusion embedding in Zorro and MCA, while in EAO it is not.[1]

When using a 3 modality dataset, the difference between EAO and Zorro models is more significant. EAO contrasts all possible pairs of unimodal and 2 modality embeddings, each generated from a separate forward pass of the same transformer block, while Zorro separately contrasts each of 3 unimodal embeddings with a single fusion embedding, all calculated in a single forward pass. MCA combines the single forward pass structure of Zorro and the 2 modality fusion representations from EAO, while also creating and contrasting fusion embeddings for all other possible modality combinations. In the next section, MCA, along with implementations of Zorro and EAO, are presented in further detail.

## 3 MODEL

This section first introduces MCA and then describes other model implementations. The core components of MCA are fusion embeddings for all possible combinations of input modalities (Figure 3a) and a block attention mask (Figure 3b) that only allows attention to occur from the corresponding modalities. This effectively creates attention channels where each channel corresponds to the fusion of a different set of modalities. The overall model architecture is presented in Figure 3a. It consists of a series of standard transformer encoder blocks with multiheaded attention, feed forward layer, layer normalization, and a cross-attention pooling layer which pools each unimodal and fu-

---

[1] An additional difference of note is that pairs of unimodal representations are contrasted in EAO, but not in Zorro. In this study, we extend Zorro to also contrast pairs of unimodal embeddings.

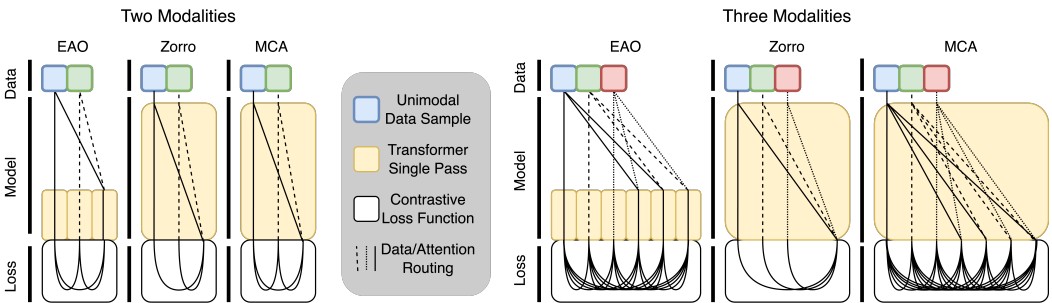

Figure 2: A comparison of the multimodal data fusion design of EAO (Shvetsova et al., 2022), Zorro (Recasens et al., 2023), and MCA (this work) where data is fused and with various combinations of modalities. The diagram demonstrates fusions for two and three modalities, demonstrating the similarities and differences between the studied models when increasing the number of modalities.

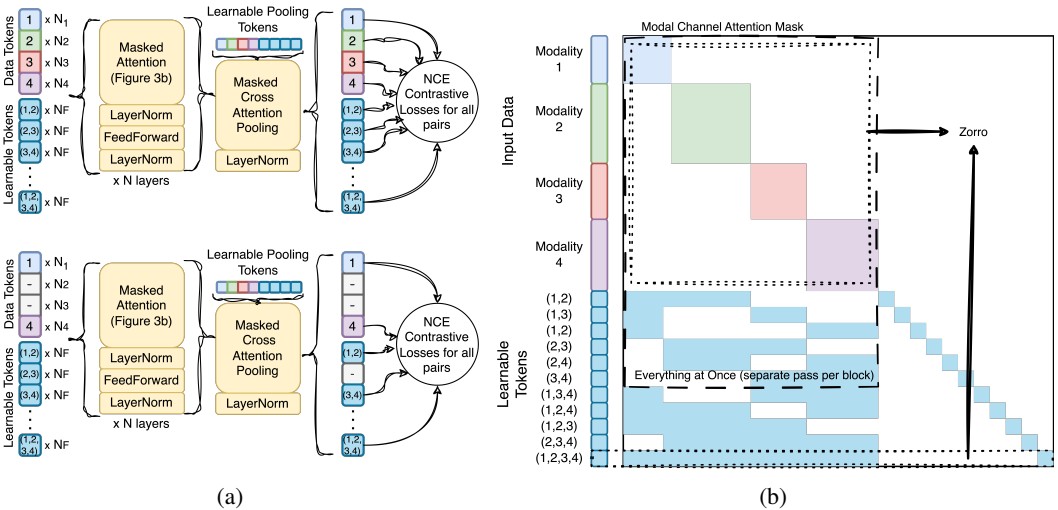

Figure 3: (a) MCA model architecture demonstrating a single forward pass for modal fusion with 4 modalities. The upper figure demonstrates when all modalities are present and the lower figure shows an example of loss masking when 2 modalities are absent. $N_i$ represents the number of tokens of the related type. (b) An example of a modal channel attention mask for a 4 modality dataset with all possible modality combinations. Inside boxes correspond to attention in other models. In EAO, no learnable fusion tokens are used and each unimodal and 2 modality fusion is performed in a separate forward pass. The attention mask used in Zorro is exactly as shown by including the learnable fusion tokens with attention from all modalities.

sion representation separately through attention masking. Noise contrastive estimation (NCE) loss is applied between these pooled embeddings. The attention block uses the attention mask shown in Figure 3b. Unimodal token blocks are only able to self-attend. Also used but not explicitly shown in this diagram are trainable transformations on the input data which are applied on a token-by-token basis in order to encode tabular data or compress pretrained frozen embeddings as inputs. Input data transformations are explicitly defined in the METHODS.

Zorro and EAO models were implemented within the MCA framework in order to experiment with their performance on sparsely multimodal data and compare them with MCA. Implementing Zorro in this framework is straightforward due to the model similarity. To do so, the attention mask was reduced to the components identified as Zorro in Figure 3b, where a single "all modality" channel is present (corresponding to $(1, 2, 3, 4)$ in the figure). In order to focus on the effects of modality fusion blocks when comparing models, we include contrastive loss between unimodal

embeddings in this Zorro implementation, as these are also present in both EAO and MCA. To implement EAO, the attention mask was removed[2] and a separate forward pass through the same transformer encoder layers was performed for each unimodal or bimodal combination of inputs. Token pooling to generate embeddings in EAO is performed as described in Shvetsova et al. (2022), where output tokens from each forward pass are averaged and then projected with a linear layer before being used as embeddings. These pooled embeddings were used to calculate the contrastive loss and gradient. While different than the cross attention pooling described for MCA and Zorro, the difference is likely minimal because no feedforward layer is applied in cross attention pooling and attention masking prevents mixing between unimodal and/or fusion representations. By combining implementations of these models, the comparison of data fusion methods are highlighted, rather than other model implementation details.

To pretrain the implemented models with sparsely multimodal data, a sample and loss masking strategy was used, described graphically in Figure 3a. Padding tokens are used for any missing modalities which are then masked from attention in the transformer encoders. This prevents the padding tokens from attending to any other tokens. While it would be possible to adjust the MCA mask (and equivalently Zorro attention mask) to remove the tokens of a missing modality from the model forward pass altogether, samples must have the same number of tokens for efficient batching and thus a padding mask strategy was chosen to allow flexibility in mixing samples with heterogeneity of modality combinations.

The resultant loss function for modal-incomplete samples was chosen to include any fusion tokens for which at least one modality is present. For example, if modality 1 exists in a sample and modality 2 is absent, a fusion token $(1, 2)$ will be attended to by only modality 1, but a loss will still be calculated between embeddings for 1 and $(1, 2)$. If both modality 1 and modality 2 are absent, the fusion token $(1, 2)$ will not be attended to by any tokens and no losses will be calculated using it. This choice of loss masking strategy was chosen such that the Zorro model's contrastive loss with a single fusion channel was preserved even with sparsely multimodal data. A variety of other possible designs for loss and token masking are conceivable, but are beyond the scope of the present study to explore.

A set of consistent hyperparameters were chosen across all models to train efficiently and fall within standard ranges of parameters for transformer encoders. This allows for focus on comparisons of multimodal fusion methods and modal sparsity. The transformer encoders use a hidden size of 512 and 8 attention heads. The feed-forward layers use a feed-forward multiplier of 4 and the GeGLU activation function. There are a total of 88 fusion tokens used in both Zorro and MCA. In Zorro, all 88 fusion tokens are pooled in the pooling layer as a single channel, while in MCA, each channel uses 8 tokens and 11 channels are present as depicted in Figure 3b where each channel of 8 tokens is shown as a single square. All models are very close in parameter count.

## 4 METHODS

### 4.1 DATASETS

#### 4.1.1 CMU-MOSEI

The CMU-MOSEI dataset was obtained and processed using the mmdatasdk Version 1.1 using the included word level alignment example. This results in 23248 samples of aligned embedded data corresponding to glove vector, OpenFace, COVAREP, and FACET encoders.(Zadeh et al., 2018) A test split is randomly chosen with 2324 samples. While raw CMU-MOSEI dataset is comprised of text, audio, and video components, these components are unavailable publicly. Instead, the processed components are used as 4 separate modalities.

For each modality, a sample is a series of embeddings for multiple time steps. The number of vectors vary per sample, due to the embedded video clips having varying duration. To prepare the modalities for input into the transformer block, each vector in a sample is transformed by using a linear layer and layer normalization, resulting in a token embedding size of $N_{emb}$. This normalized, transformed vector is then added to a standard sinusoidal positional embedding vector to encode it's position in the sequence of vectors for a given modality. No other learnable token embedding is used for these

---

[2]Attention masking was applied for padding tokens, but not for modal attention

samples. The result of this process is that each sample token is transformed into a token embedding for input into the models studied in this paper

### 4.1.2 TCGA

The Cancer Genome Atlas (TCGA)(Weinstein et al., 2013) provides a multi-omics dataset that consisting of tabular data for gene expression, reverse phase protein arrays (RPPA), DNA methylation, and miRNA measurements. This data was downloaded from the supporting information of Weinstein et al. (2013). To reduce the number of gene expression and DNA methylation columns in the dataset, the top 800 genes and methylation sites with the highest variance were used to create a signature of the gene expression and methylation data. For RPPA data and miRNA tables, there are 198 protein columns and 662 MiRNA columns. To align unimodal samples into a multimodal dataset, we use provided identification numbers for patient and sample, resulting in an intersection of 7017 samples that have all modalities. A test split is randomly chosen with 707 samples.

To encode TCGA tabular data, values are passed token-wise through a trainable 2 layer $\text{MLP}(1, N_{emb}, N_{emb})$ with ReLU activation function. This allows a continuous representation of the tabular value into a vector with the same size as the transformer encoder embedding space $N_{emb}$. The tabular column index is encoded with a standard learnable embedding vector of size $N_{emb}$ for each index. The value and column index encodings are added together to form the input token embedding vectors for the transformer encoders.

### 4.2 MODAL SPARSITY

In order to evaluate the performance of EAO, Zorro, and MCA with missing modalities, modality data is dropped from samples in the datasets. This procedure is performed prior to training, such that the same data is used consistently. For each modality in each sample, the probability that it is dropped is equivalent to the modal sparsity. The modal sparsity ($S$) reported in the following figures thus represents the fraction of dropped samples in each modality.

$$S = \frac{1}{N_S} \sum_{i=1}^{N_S} M_i / M_T \tag{1}$$

where $N_S$ is the number of samples in the dataset, $M_i$ is the number of modalities in a sample $i$ and $M_T$ is the number of possible modalities in the dataset. Due to the training cost of many models, experiments were performed with datasets constructed to have 0, 0.2, 0.4, 0.6, and 0.8 modal sparsity. Throughout this study, the relation between $S$ and model performance is explored, resulting in figures which demonstrate metrics as a function of $S$

Since modalities in each sample are dropped with equal probability and datasets used in this study contain 4 modalities, a modal sparsity of 0.2 indicates that most samples have 3 modalities present while at a modal sparsity of 0.6 indicates that most samples have only 1 or 2 modalities present. As described above, when a modality is dropped from a sample, a padding token is used for all tokens corresponding to that modality such that they are masked from subsequent attention blocks and subsequent loss function terms are dropped as described in MODEL.

### 4.3 TRAINING

Training was performed for MCA and Zorro models on 4 A10G Nvidia GPUs with an allocated memory requirement of 17GB). Due to the additional memory requirements of EAO (41GB), training was performed on 4 A100 GPUs. All training runs used a distributed data parallel strategy, with all embeddings collectively used for loss function calculations. Training hyperparameters were chosen identically for Zorro, EAO, and MCA experiments. An effective batch size of 32 (8 samples per GPU) and cosine scheduled learning rate with a maximum of $10^{-4}$ and warm up of 2000 steps were used for all experiments. Test splits of datasets were selected randomly as a fraction of 0.15 of a dataset and used subsequently for downstream tasks. For CMU-MOSEI experiments, 32 epochs are trained. For TCGA fusion, 128 epochs are trained. The epoch number selected for evaluating embeddings was hand selected by identifying the best set of test loss scores for each model/dataset

pair at all modal sparsities as demonstrated in the Appendix. The model and training code were developed using Pytorch.(Paszke et al., 2019)

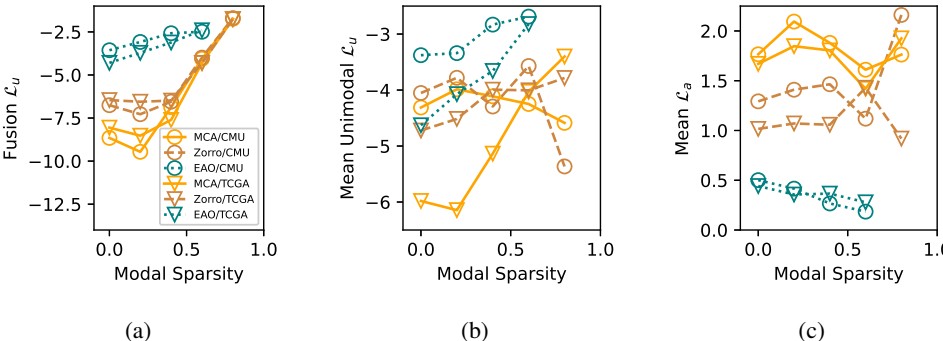

(a)                                   (b)                                   (c)

Figure 4: Uniformity and alignment metrics as a function of dataset sparsity for CMU-MOSEI and TCGA dataset embeddings calculated from test dataset splits for (a) uniformity of fusion embeddings ($\downarrow$); (b) mean uniformity of unimodal embeddings ($\downarrow$); (c) Mean alignment between unimodal and fusion token embedding spaces.($\downarrow$);

## 5 RESULTS

### 5.1 UNIFORMITY AND ALIGNMENT

This section aims to analyze the characteristics of the embeddings produced by the trained models. Embedding spaces trained with contrastive learning are characterized by alignment ($\mathcal{L}_a$) and uniformity ($\mathcal{L}_u$).(Wang & Isola, 2020) These are defined by

$$\mathcal{L}_a = \mathbb{E}_{x,y}[||f(x) - f(y)||_2^2] \tag{2}$$

$$\mathcal{L}_u = \mathbb{E}_{x,y}[e^{-2||f(x)-f(y)||_2^2}] \tag{3}$$

where $\mathbb{E}_{x,y}$ is the expectation value over variables $x$ and $y$. For $\mathcal{L}_a$, $x, y$ are positive pairs from two different embedding types (e.g. a fusion and unimodal embeddings), while for $\mathcal{L}_u$, $x, y$ are pairs of embeddings from a single embedding type.

To compare the $\mathcal{L}_u$ and $\mathcal{L}_a$ of generated embedding spaces across models and varied modal sparsity we calculate these metrics using the test splits of both TCGA and CMU-MOSEI datasets. While there are multiple fusion embeddings in MCA, to compare with Zorro and EAO, we use the fusion embedding that includes attention from all modalities in the following analyses. When necessary, we take the mean of the metric calculated for each unimodal embedding. For example, the mean unimodal $\mathcal{L}_u$ shown in Figure 4b is the mean of the $\mathcal{L}_u$ after calculating it separately for each unimodal embedding type.

If a model is trained to generate an embedding space with lower $\mathcal{L}_u$ (i.e. better, indicating a more uniformly distributed embedding space), the $\mathcal{L}_a$ of positive (matching) embeddings will tend to be increased (i.e. worsened, indicating less alignment).(Wang & Isola, 2020) The $\mathcal{L}_u$ and $\mathcal{L}_a$ of MCA and Zorro have no clear trend up to a modal sparsity of 0.4, while EAO demonstrates an monotonic increase in $\mathcal{L}_u$ and decrease in $\mathcal{L}_a$. As the modal sparsity is increased beyond 0.4, both EAO and Zorro models have worsened $\mathcal{L}_u$, even though $\mathcal{L}_a$ is not as strongly affected. For the smaller dataset (TCGA), $\mathcal{L}_u$ increases in MCA with increasing modal sparsity.

EAO has better $\mathcal{L}_a$ than MCA and Zorro. This is likely because the fusion embedding of EAO is constructed directly from the average of unimodal embeddings and it's effect is apparent in ranking and recall metrics. Correspondingly, the $\mathcal{L}_u$ of EAO is significantly worse than MCA and Zorro. This could be because EAO does not have a mechanism that fuses all unimodal representations at once, where attention is calculated with, at most, only 2 modalities at a time. MCA demonstrates better $\mathcal{L}_u$ and worse $\mathcal{L}_a$ of both unimodal and fusion embeddings than Zorro. Regardless, this increase in $\mathcal{L}_u$ tends to outweigh the effect of $\mathcal{L}_a$ on ranking and recall metrics when comparing these two models, as described in the next section.

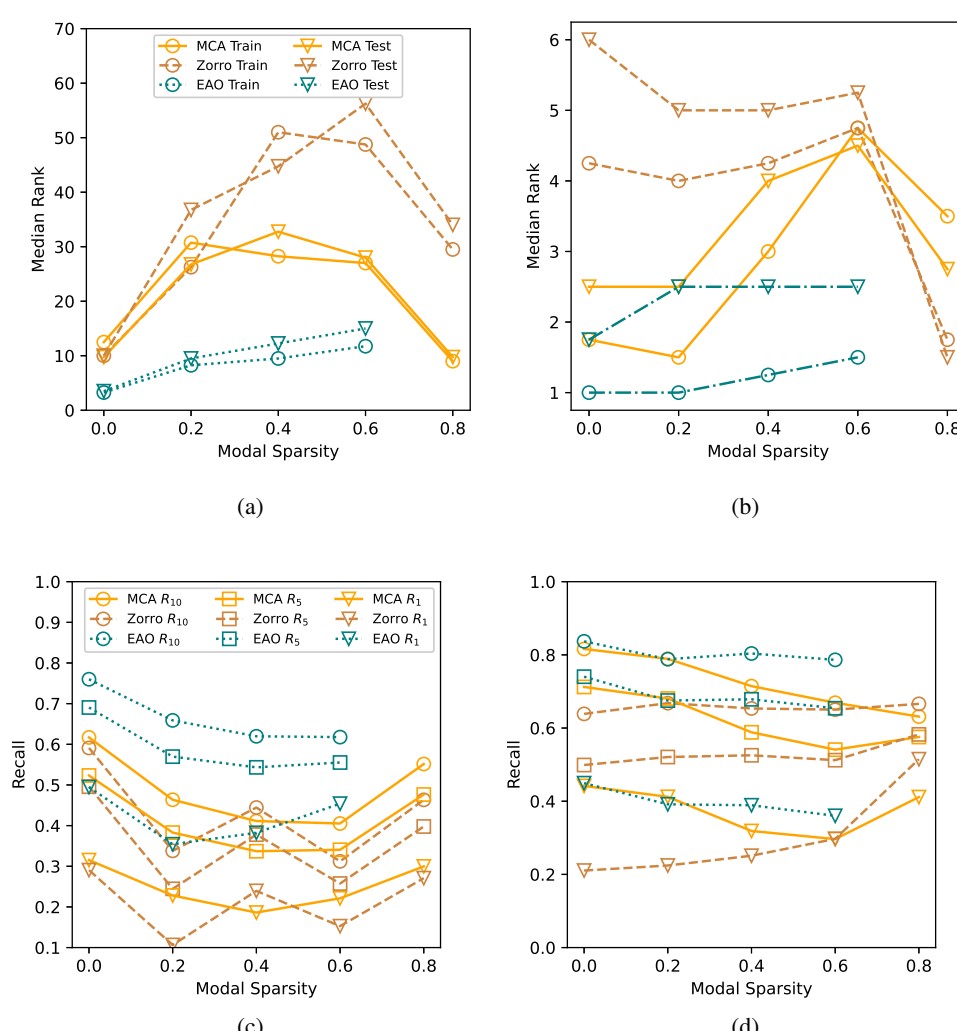

Figure 5: Rank and recall metrics for embeddings from models trained with various modal sparsity on the CMU-MOSEI and TCGA datasets. (a) Median Rank for CMU-MOSEI (↓); (b) Median Rank for TCGA (↓); (c) Recall for CMU-MOSEI (↑); (d) Recall for TCGA (↑);

## 5.2 RANKING AND RECALL

Ranking and recall metrics were examined for the generated test splits of the datasets (median rank, $R_1$, $R_5$, and $R_{10}$). These recall metrics demonstrate the ability to use a unimodal embedding to recall it's matching fusion embedding. Ideally, any matching pair of unimodal and fusion embeddings has a greater similarity than all non-matching pairs. The cosine similarities of a unimodal embedding to each fusion embedding is used to calculate the rank. The median rank is the median value of these ranks (Figures 5a and 5b). The recall ($R_x$) is equivalent to the probability that the correct fusion embedding is in the top $x$ most similar fusion embeddings (Figures 5c and 5d. Note that as modal sparsity is increased, the size of the dataset is reduced by samples which have all modalities dropped, which may affect the results.

EAO has the best performance in ranking methods, which is not surprising given that the fusion embeddings are calculated as the average of unimodal and 2 modality embeddings. In the TCGA, MCA performs nearly as well as EAO, even with a modal sparsity of 0.2. The median rank is improved in MCA over Zorro in most cases. In the larger CMU-MOSEI dataset the difference is greatest at high modal sparsity, while in the smaller TCGA dataset it is greatest at low modal sparsity.

This may be due to better uniformity and worse alignment in MCA. At the highest sparsity of 0.8 the median rank drops significantly. This may be due to a fewer number of embeddings overall.

Recall shows similar trends to median rank. EAO demonstrates the best performance overall, which is most pronounced in the larger dataset. MCA has better recall than Zorro in the majority of experiments. At 0.2 modal sparsity in the smaller dataset, MCA has improved recall to EAO. These results show the improvement of MCA over Zorro as a method of building an embedding space for multimodal retrieval, even when only a single modality is available at inference time and training examples are sparsely multimodal. Furthermore, the improved alignment of EAO embeddings clearly leads to better ranking and recall metrics.

## 5.3 REGRESSION AND CLASSIFICATION

The analysis of the generated embeddings on downstream regression and classification tasks is a crucial component of validating model performance. In this section the linear probing performance of embeddings as a function of modal sparsity is explored. A linear layer is trained using embeddings produced from the dataset training split using either L1 loss for regression or cross entropy loss for

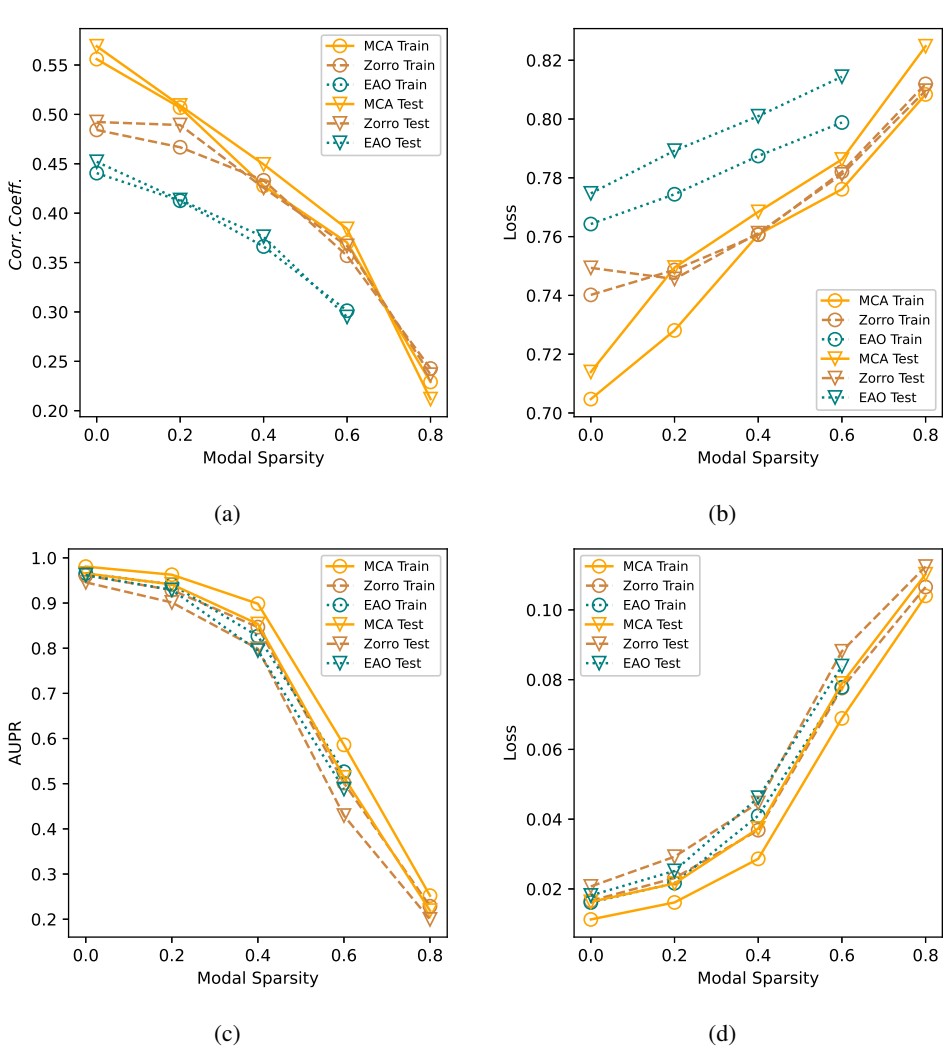

Figure 6: Comparison of performance on linear probing tasks of generated embedding spaces. (a) Correlation between true and predicted values for CMU-MOSEI sentiment regression task (↓); (b) Loss calcualted for CMU-MOSEI regression task (↓) (c) Average AUPR value for multi-class classification of TCGA tumor type (↑); (d) Loss calculated for TCGA classification task (↓);

classification. The metrics displayed in Figure 6 are calculated using the trained linear model to run inference on test dataset splits which have not been used to train the embeddings or the linear probe. Importantly, no pretrained model weights are relaxed in this evaluation. All tasks are thus a direct linear probing of the generated embedding spaces, trained on the training data split and tested on the test data splits. In general MCA provides improved results over EAO and Zorro for both datasets.

The CMU-MOSEI sentiment analysis task is a regression to a single value. This value ranges between 0 and 1, corresponding to negative and positive sentiment. In the initial study describing the CMU-MOSEI dataset, a correlation coefficient of 0.54 is achieved using an LSTM-based modal fusion architecture.(Zadeh et al., 2018) Results are presented for this task in Figure 6a. MCA meets this baseline result with only linear regression of the produced embeddings when no modal sparsity is present, while Zorro and EAO do not. As modal sparsity is increased, the correlation coefficient is reduced. MCA maintains improvement over Zorro for the test data correlation coefficient up to a modal sparsity of 0.6. However, EAO has significantly lower correlation coefficient for this task than both MCA and Zorro. This suggests that high fusion embedding uniformity is beneficial for this task.

The task performed for the TCGA dataset is a multiclass classification problem with 32 classes. These correspond to the type of cancer present in the specimen from which a sample was generated. The class-averaged area under the precision-recall curve (AUPR) is presented in Figure 6a. All models perform well at this task up to a modal sparsity of 0.4, after which the performance drops significantly. MCA has the best performance in all experiments, but unlike the CMU-MOSEI regression task, EAO has better performance than Zorro. It is surprising that EAO performs better than Zorro on this task. This may be because the task itself is comparatively simple and information is required from only a subset of the modalities. This points to the benefit of contrasting all possible combinations of modalities in the MCA model in order to create embeddings that have good performance on a wide range of tasks.

## 6 CONCLUSION

This study investigates the performance of multimodal embedding techniques in scenarios with varying degrees of modal sparsity. By examining MCA, Zorro, and EAO embeddings on both ranking/recall metrics and downstream regression/classification tasks, advantages and trade-offs associated with each method are demonstrated. MCA consistently outperformed Zorro across most experiments. Its ability to contrast all combinations of modalities enables it to generate embeddings that maintain improved uniformity, leading to improved results in both ranking-based retrieval tasks and downstream linear probing evaluations. While EAO excelled in ranking tasks due to its post-inference calculation of fusion embeddings, it performed worse than MCA in regression/classification tasks where complex multimodal interactions are required.

Overall, the results show the potential of MCA as a method for generating robust multimodal fusion embeddings, particularly in sparsely multimodal datasets. Its demonstrated improvements over Zorro and EAO suggest that incorporating fine-grained contrastive strategies into embedding models can significantly improve model performance. These findings pave the way for future research into advanced multimodal data fusion techniques with applications in multimodal retrieval and downstream tasks.

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

# A   APPENDIX

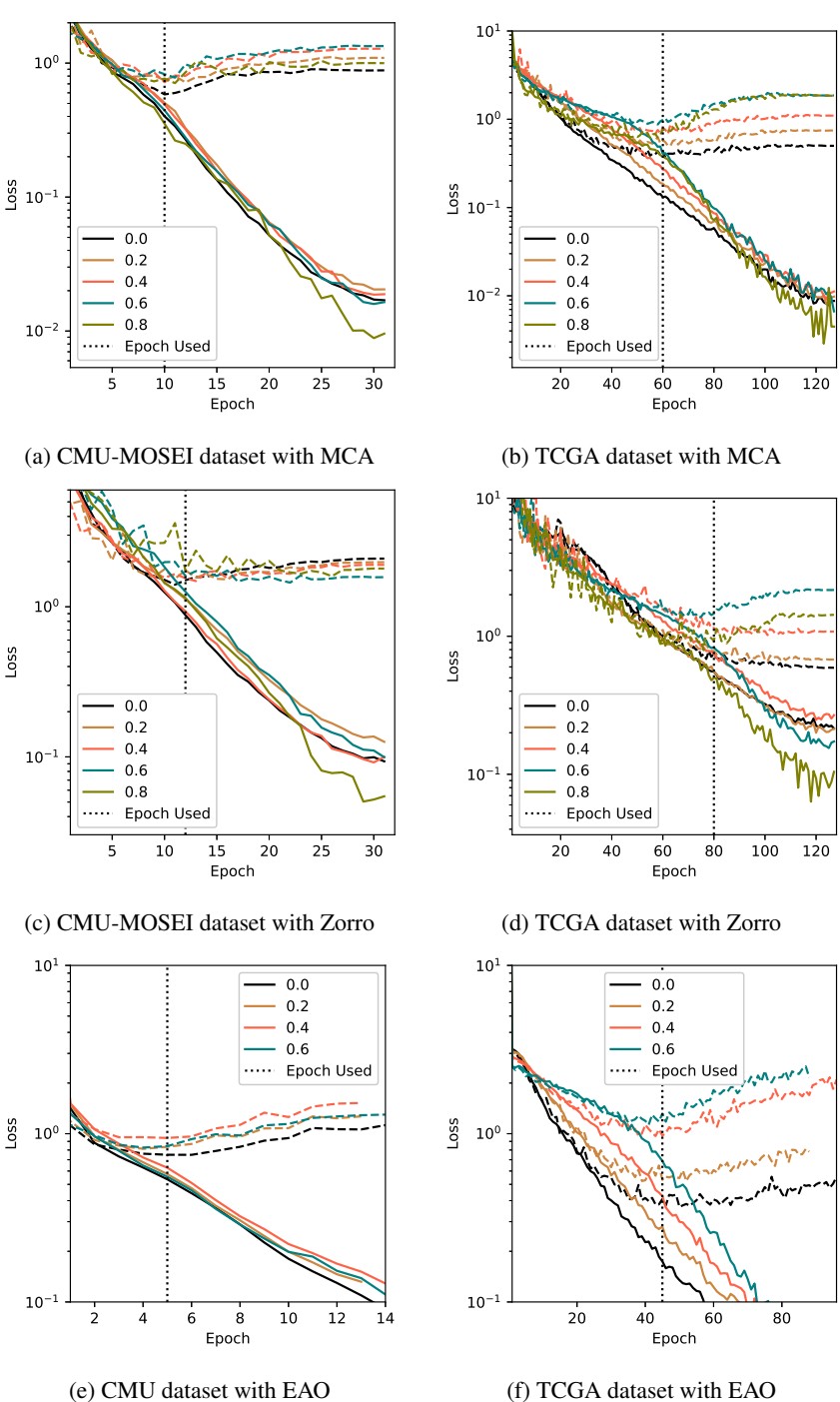

(a) CMU-MOSEI dataset with MCA

(b) TCGA dataset with MCA

(c) CMU-MOSEI dataset with Zorro

(d) TCGA dataset with Zorro

(e) CMU dataset with EAO

(f) TCGA dataset with EAO

Figure 7: Epoch averaged losses at various modal sparsities (legend) for train (solid lines) and test (dashed lines) splits.

