# OpenReview forum: "Sparsely multimodal data fusion"
_ICLR.cc/2025/Conference — Submitted to ICLR 2025_

### Official Review · Reviewer_fo1E · 2024-10-30

**Soundness:** 2
**Presentation:** 2
**Contribution:** 1
**Rating:** 5
**Confidence:** 4

**Summary:**

This paper presents a sparse multimodal fusion method in the presence of modal incompleteness. An extension to the masked multimodal transformer model is proposed which incorporates modal-incomplete channels in the multi-head attention mechanism called modal channel attention (MCA). The validity and usefulness of the method is verified by experiments on two multimodal datasets with different metrics and scenarios.

**Strengths:**

* This paper demonstrates the importance of how to cope with representation learning in the case of incomplete modalities in real applications.

* The authors provide extensive implementation details and interesting experiments to demonstrate the need for the proposed structure.

* The authors' review of the relevant literature is well organized.

**Weaknesses:**

* This paper is clearly not ready for publication, including confusing typography, inadequate page counts for major sections, and formatting errors. For example, Fig and Figure are mixed up and line 221 is missing punctuation.

* The experimental section focuses on the comparative analysis of different metrics. However, the authors lacked an effective comparison with previous work to highlight the advantages of the proposed methodology.

* In addition, this study does not have enough qualitative analysis and ablation studies to explore the effect of different combinations of modal deletions in the proposed structure on performance.

**Questions:**

* As far as I know, CMU-MOSEI primarily provides information in three modalities, including audio, video, and text. I am not sure what exactly the four modalities the authors are referring to when they use them.

* The core structure presented in this paper is very similar to previous work [1], leading to limited improvements and contributions. I suggest that the authors clarify the differences between them in detail to highlight their contribution.

[1] Shvetsova, N., Chen, B., Rouditchenko, A., Thomas, S., Kingsbury, B., Feris, R. S., ... & Kuehne, H. (2022). Everything at once-multi-modal fusion transformer for video retrieval. In Proceedings of the ieee/cvf conference on computer vision and pattern recognition (pp. 20020-20029).

---

> ### Author Response · Authors · 2024-11-22
>
> Dear reviewer,
> Thank you for taking the time to review this manuscript and provide helpful comments and questions, which have been used to improve the manuscript both in overall readability and add improved detail to the technical content. We hope you will be able to review the revised uploaded PDF of the manuscript and welcome any additional comments and questions that you have. The present comments and questions are addressed point-by-point below.
>
> --The experimental section focuses on the comparative analysis of different metrics. However, the authors lacked an effective comparison with previous work to highlight the advantages of the proposed methodology.
>
> This is an excellent point and existing previous works which are functional with more than 3 modalities and with samples that are missing modalities are quite few. We have added the following information on our selection of models for comparison:
> "None of these aforementioned models [Note, these are previous works described in the text] generate a fusion embedding space, which is required for tasks based on embeddings like retrieval and linear probing for regression and classification. There is no clear extension to the aforementioned models where a fusion embedding is close to it's unimodal embeddings and to embeddings generated from modal-incomplete samples. Since the goal of this study is to explore fused embedding spaces for data with more than 2 modalities when the datasets are sparsely multimodal, we now turn to models designed for related purposes. "
>
> To address this, the manuscript has been restructured to provide a comparative analysis between MCA and two existing models, Zorro and Everything at Once, where the latter models have been extended to apply to 4 modality datasets. Note that MMA in the original text was the previously published Zorro model generalized to additional modalities, and additional details on this in the text for clarity.
>
> --In addition, this study does not have enough qualitative analysis and ablation studies to explore the effect of different combinations of modal deletions in the proposed structure on performance.
>
> This is an excellent point and we hope to further explore this avenue in the future. However, this would require training many such models with fully ablated modalities and is beyond the scope of this manuscript. This will be important to explore in the future, especially with larger multimodal datasets, but we hope that the revised restructuring of the manuscript to demonstrate MCA, Zorro, and Everything at Once in the context of sparsely multimodal data may improve the value of the qualitative analysis otherwise.
>
> --As far as I know, CMU-MOSEI primarily provides information in three modalities, including audio, video, and text. I am not sure what exactly the four modalities the authors are referring to when they use them.
>
> We've added some additional clarity on this in the text in the Datasets section
> "This results in 23248 samples of aligned embedded data corresponding to glove vector, OpenFace, COVAREP, and FACET encoders.\citep{zadeh2018multi}  A test split is randomly chosen with 2324 samples. While raw CMU-MOSEI dataset is comprised of text, audio, and video components, these components are unavailable publicly. Instead, the processed components are used as 4 separate modalities."
> These are time-aligned data structures for each video as described in Zadeh, A. B. et al. - glove vectors are word-level embeddings from video transcripts, FACET are 6 basic emotions predicted from facial expressions, OpenFace are embeddings related to face and pose, and COVAREP is an audio spectrogram embedding.
>
> --The core structure presented in this paper is very similar to previous work [1], leading to limited improvements and contributions. I suggest that the authors clarify the differences between them in detail to highlight their contribution.
>
> In order to address this, we have restructured the manuscript to highlight the comparison between MCA and existing models Zorro and Everything at Once (EAO), where the latter two have been extended to embed datasets with more than 3 modalities and use sparsely multimodal data. Particularly in Figure 2, we show how MCA and Zorro compare with EAO in terms of the flow of modality data and in Figure 3b, we show how MCA is designed to include the strengths of both Zorro and EAO. Finally, in our empirical analysis, we show that although MCA generates embeddings which are better for recall than Zorro, it is still not as good as EAO. However, we also show that EAO embeddings underperform MCA embeddings on downstream tasks, we surmise that this is due to additional modality fusion channels in the MCA method. We also note that MCA and Zorro require significantly less GPU memory than EAO to train.

---

### Official Review · Reviewer_h5n8 · 2024-10-31

**Soundness:** 3
**Presentation:** 3
**Contribution:** 3
**Rating:** 6
**Confidence:** 4

**Summary:**

Traditional multimodal learning methods often assume that all modes are uniformly present in training and reasoning, which may be impractical in practical applications. To solve this problem, the authors propose a modal channel attention (MCA) model, which is an extension of the masked multimodal attention (MMA) approach. By integrating a dedicated attention mechanism, MCA achieves efficient fusion and embedding of multimodal data, even in the case of sparse modes. We evaluate MCA on the CMU-MOSEI and TCGA datasets, and the results show that MCA outperforms MMA in creating aligned, uniform embeddings with different modal sparsity. In downstream tasks such as multimodal sentiment analysis and tumor type prediction, MCA also shows higher recall rates and classification/regression performance, making it a versatile tool for real-world multimodal data fusion.

**Strengths:**

1. MCA provides a novel method for dealing with modal sparsity in multimodal transformers, addressing a common limitation in multimodal learning. The approach enhances the MMA model by adding a modality-aware attention channel, innovatively allowing robust embedding creation from incomplete datasets.
2. The study is rigorous, using two datasets (CMU-MOSEI and TCGA) to evaluate the model's effectiveness across varied tasks, including sentiment analysis and cancer classification. The experiments are thorough, with evaluations under multiple levels of modal sparsity and comparisons with established models.
3. This model has important implications for real-world applications where complete multimodal data is often unavailable, such as in medical data analysis and surveillance systems. MCA enables more resilient model applications by tolerating missing modalities without significant loss of performance, thus broadening the scope of multimodal transformers.

**Weaknesses:**

1. This paper does not explore in depth the interactions between modes to further explore the specific impact of different combinations of modes on MCA model performance. You may wish to consider exploring whether particular combinations are more conducive to improving model performance in order to gain a fuller understanding of the relative importance of each mode.
2. Although MCA is described as an efficient solution, this paper does not further quantify its computational efficiency and resource consumption on larger or more complex datasets to effectively assess the feasibility of its practical application.
3. The attention mechanism of the MCA model in dealing with missing modalities is still complex and not easy to understand intuitively. I would like to see the interpretability of the model in applications, which would help build user trust in the model's decisions.
4. The comparison experiments in this paper are limited to comparisons with the MMA model, which makes it difficult to fully demonstrate the relative strengths of the MCA model in multimodal approaches. You can evaluate the advantages and limitations of the MCA model more comprehensively by increasing the number of comparison experiments with multimodal methods other than MMA.

**Questions:**

1.	In the MCA model, are important modalities treated differently from minor modalities? Or are different modalities given different weights in the attention mechanism?
2.	At which critical point does the performance of MCA significantly degrade as modal sparsity increases? Are there optimisation methods to maintain embedding quality at higher sparsity?

---

> ### Author Response · Authors · 2024-11-22
>
> Dear Reviewer,
> Thank you for your insightful suggestions. We have taken them into account to prepare a revised manuscript and hope you will be able to review the uploaded PDF. It includes general improvements to readability, additional high-resolution diagrams, and extended details of the architecture, design choices, and analysis of the results. Note that we have renamed the MMA model to Zorro, as it is effectively the model used in this prior publication (as was described in original submission). The new submission also includes additional comparative analysis between existing models Zorro and Everything at Once in order to provide insights into the comparative performance between models. Your comments and questions are addressed one-by-one below:
>
> --You may wish to consider exploring whether particular combinations are more conducive to improving model performance in order to gain a fuller understanding of the relative importance of each mode.
>
> This is a direction of research which could be performed in a future publication, especially on larger datasets. It would require the analysis of additionally trained models with structured ablations, which are out of the scope to perform for this study. We have instead included training of the Everything at Once (EAO) model in order to include further comparative analysis with the MCA and Zorro models. We have used the comparisons of these architectures to provide insights into the relative performance of these models.
>
> --Although MCA is described as an efficient solution, this paper does not further quantify its computational efficiency and resource consumption on larger or more complex datasets to effectively assess the feasibility of its practical application.
>
> While training on larger and/or more complex datasets is out of the scope of possibility for this submission, we clarify in the publication that the efficiency of MCA by mechanism of multiple embeddings generated in a single forward pass compared with EAO, with generates them one-by-one. We note in the revision that this results in a required memory allocation for training of 17GB for MCA and Zorro, but 41GB for EAO and also clarify the differences between the organization of these embeddings with respect to loss function in Figure 2.
>
> --The attention mechanism of the MCA model in dealing with missing modalities is still complex and not easy to understand intuitively. I would like to see the interpretability of the model in applications, which would help build user trust in the model's decisions.
>
> Interpretability would be an excellent addition to a comprehensive analysis on a large dataset with additional downstream tasks and could be part of a later publication. In the limited space of this conference publication this study focuses on comparative analysis of multimodal fusion methods which are functional with sparsely multimodal data and the introduction of MCA. We have instead included additional information on model architectures and their comparisons. This includes information on the implemented loss masking strategy and the padding strategy used when modalities are ablated from a data sample. This is the 3rd and 4th paragraphs of the Model section of the revised manuscript, which are too long to paste into this reply. This information should provide some clarity on how the model deals with missing modalities, although further analyses should include attention weight analysis, interpretability, and further structured ablations.
>
>
> --The comparison experiments in this paper are limited to comparisons with the MMA model, which makes it difficult to fully demonstrate the relative strengths of the MCA model in multimodal approaches. You can evaluate the advantages and limitations of the MCA model more comprehensively by increasing the number of comparison experiments with multimodal methods other than MMA.
>
> This is a valuable suggestion and our manuscript revision has attempted to focus on addressing it. We have restructured the manuscript to describe in detail the differences between MCA, Zorro and EAO models, provide insightful empirical analyses on their performance on a range of important tasks when modal sparsity is changed, and determine architectural differences which contribute to the differences in performance. The results and conclusions of the paper have not changed, despite adding additional results for comparing with EAO, and the added reasoning does indeed describe the advantages and limitations.

---

> ### Author Response · Authors · 2024-11-22
>
> --In the MCA model, are important modalities treated differently from minor modalities? Or are different modalities given different weights in the attention mechanism?
>
> Since the attention mask in MCA creates channels for all combinations of modalities, including unimodal channels, and all embeddings are contrasted pair-wise, it should prevent modalities from having different weight in the fusion channel that includes all modalities at once, since it is trained to match all unimodal channels, all bimodal channels, trimodal channels, etc. However, were a modality to have vary little variance overall, it may not be able to contribute to the training of the model weights and then it's specific loss components should also have relatively low performance. This question could be further addressed in a future publication.
>
> --At which critical point does the performance of MCA significantly degrade as modal sparsity increases? Are there optimisation methods to maintain embedding quality at higher sparsity?
>
> There seems to be a change after modal sparsity of 0.4 where fusion uniformity becomes significantly worse in MCA, while the alignment becomes worse after 0.2. Many additional datapoints will be required to gain further insights into these trends which are the goal of a future study. In terms of task performance, these tend to get worse with any modal sparsity. In classification using the smaller TCGA dataset, after 0.4 modal sparsity performance decreases more rapidly. We are not aware of any optimization techniques (other than the strategy of loss masking employed here) that can be used to improve embedding quality under sparse multimodal conditions.

---

> > ### Comment · Reviewer_h5n8 · 2024-11-23
> >
> > Thanks for your reply. I hope you can provide some detailed comparison results to support the effectiveness of MCA

---

> > > ### Author Response · Authors · 2024-11-23
> > >
> > > Yes. We have provided detailed comparison of MCA with Zorro and Everything at Once models in the revised submission. Note that there are no other available models that can be used to contrastively pretrain fusion embeddings on multimodal data with 4 modalities, that we are aware of. Please let us know if any other specific models should be compared.

---

### Official Review · Reviewer_927z · 2024-11-01

**Soundness:** 3
**Presentation:** 2
**Contribution:** 3
**Rating:** 5
**Confidence:** 4

**Summary:**

**Summary of this paper:**

This work proposes an extension to the masked multimodal transformer model which incorporates modal-incomplete channels in the multihead attention mechanism called modal channel attention (MCA).

**Strengths:**

1.	This paper addresses an interesting problem by extending the masked multimodal attention model architecture to work with modality-incomplete datasets, a common scenario in real-world applications.

2.	The paper is straightforward.

3.	Experimental results demonstrate the effectiveness of the proposed method.

**Weakness:**

There is no visual comparison between the proposed architecture and previous structures, which would help clarify the improvements and differences.

**Comments, Suggestions And Typos:**

1.	It is recommended to add a figure in the introduction to briefly introduce the task of this paper, or to add a figure somewhere to compare the proposed model architecture with previous MMA architecture.

2.	Figure 1 is not good-looking, the authors are encouraged to improve the quality of the figure.

3.	For clarity and rigor, it is recommended to express important definitions or functions (i.e., modal sparsity) by formulas.

4.	The citation formats for figures differ in this paper.

**Strengths:**

Refer to the summary

**Weaknesses:**

Refer to the summary

**Questions:**

Refer to the summary

---

> ### Author Response · Authors · 2024-11-22
>
> Dear reviewer,
>
> Thank you for review and insightful suggestions, which have been used to improve the manuscript and will be addressed in detail below. A revised PDF file has been uploaded with improved overall writing clarity, additional high resolution diagrams, and additional comparative analysis. Note that we have renamed MMA in the manuscript to Zorro, since MMA is effectively the model implemented in the referenced Zorro paper. We hope that you will be able to review the improved manuscript and provide a revised rating.
>
> --There is no visual comparison between the proposed architecture and previous structures, which would help clarify the improvements and differences.
> We have created a comparative diagram that compares the principles of MCA, Zorro, and Everything at Once (EAO) models side-by-side for two and three modality cases. We have also included annotations to the attention mask diagram showing how Zorro and EAO are related to MCA's attention mask.
>
> --It is recommended to add a figure in the introduction to briefly introduce the task of this paper, or to add a figure somewhere to compare the proposed model architecture with previous MMA architecture.
> We have added a figure in the introduction of the paper detailing the motivation and main purposes of the developed MCA model and comparison to Zorro and EAO models. This figure shows fusion embeddings generated from sparsely multimodal data to be used in ranking/recall and downstream regression/classification tasks with modality-incomplete data.
>
> --Figure 1 is not good-looking, the authors are encouraged to improve the quality of the figure.
> We have revised this figure and created a high-resolution version which is now in the revised PDF. We apologize for the low figure quality.
>
> --For clarity and rigor, it is recommended to express important definitions or functions (i.e., modal sparsity) by formulas.
> We have included equations for modal sparsity, uniformity, and alignment.
>
> --The citation formats for figures differ in this paper.
> We have revised the citation format to be consistent throughout, with subfigures referenced in the main figure citation and the use of the format "Figure X" in the main text.
>
> We look forward to any additional comments or suggestions you may have regarding the submission.

---

### Official Review · Reviewer_FYR5 · 2024-11-02

**Soundness:** 1
**Presentation:** 1
**Contribution:** 3
**Rating:** 3
**Confidence:** 3

**Summary:**

This paper is not ready for publication in ICLR

**Strengths:**

The topic is very interesting and important.

**Weaknesses:**

The presentation is poor.

**Questions:**

No

---

> ### Author Response · Authors · 2024-11-22
>
> Dear reviewer,
>
> Thank you for your consideration of this manuscript. A revised submission has been uploaded with additional details and new high resolution diagrams. Revisions to the text of the manuscript were made to add clarity and improved readability.
>
> We hope that you are able to review the improved text and provide additional feedback and a revised rating.

---

### Official Review · Reviewer_NvQ1 · 2024-11-02

**Soundness:** 3
**Presentation:** 2
**Contribution:** 3
**Rating:** 5
**Confidence:** 4

**Summary:**

The paper proposes a method called Modal Channel Attention (MCA) to enhance masked multimodal transformer models for learning from multimodal data, even in the presence of missing modalities. The approach extends existing masked multimodal attention (MMA) by incorporating channels for incomplete modalities, which significantly improves the quality of learned embeddings.

**Strengths:**

1. It addresses the challenge of missing modalities.

2. It extends the masked multimodal attention (MMA), improving its capability to handle incomplete data.

3. Improves the quality of learned embeddings.

**Weaknesses:**

1. The contributions primarily emphasize performance improvements and extensions to the existing MMA. More detailed explanations of the specific contributions and the importance of the novelty would be beneficial.

2. The technical section is quite brief (half a page), and it would be helpful if the figure showing the architecture and self-attention provided more detail. Moreover, part of the technical section is devoted to describing the architecture by listing details like activations, hidden size, and the number of attention heads. However, no explicit motivations are provided for these choices. Additionally, the figures should have been more curated to avoid text pixelation. Given the current shape of the paper, the novelties seem to be present, but the paper fails to properly highlight them, relying mostly on an empirical evaluation.

**Questions:**

The paper suggests novel contributions, but they are not sufficiently highlighted. Can you expand on how MCA distinguishes itself from existing methods in terms of theoretical innovations or novel mechanisms, beyond empirical performance? Providing explicit motivations for architectural choices would also make the contributions clearer and more impactful.

---

> ### Author Response · Authors · 2024-11-22
>
> Thank you for your helpful comments and questions.
>
> --The paper suggests novel contributions, but they are not sufficiently highlighted. Can you expand on how MCA distinguishes itself from existing methods in terms of theoretical innovations or novel mechanisms, beyond empirical performance? Providing explicit motivations for architectural choices would also make the contributions clearer and more impactful.
>
> A revised manuscript has been uploaded which details how MCA is related to Zorro and Everything at Once (EAO) models, as well as explains the justifications for architectural choices. The MMA model has been renamed to Zorro, as MMA is effectively the Zorro model from a referenced publication, in order to better highlight that this is a comparison to an existing model. We have included improved high resolution diagrams showing these components and comparisons, and hope you are able to review the revisions. The conclusions of the manuscript have not changed, but more details are included.
>
> For example:
> "When using a 3 modality dataset, the difference between EAO and Zorro models is more significant. EAO contrasts all possible pairs of unimodal and 2 modality embeddings, each generated from a separate forward pass of the same transformer block, while Zorro separately contrasts each of 3 unimodal embeddings with a single fusion embedding, all calculated in a single forward pass.  MCA combines the single forward pass structure of Zorro and the 2 modality fusion representations from EAO, while also creating and contrasting fusion embeddings for all other possible modality combinations."
>
> Details of loss masking strategy for sparsely multimodal data and comparative reasoning for the performances of the studied models.

---

### Meta-Review · Area_Chair_aAEV · 2024-12-16

**Metareview:**

It is clear to see that all the reviewers have concerns about the technical and experimental details of this paper, and all of them voted a low score for this paper, that is, one rejection, three weak rejections, yet one weak acceptance. Based on the overall scores of the whole ICLR papers, this paper will NOT be accepted for ICLR.

**Additional Comments On Reviewer Discussion:**

All the reviewers voted a low score for this paper, and the provided rebuttal did not persuade the reviewers to change their minds about this work. This paper will be rejected without further consideration.

---

### Decision · Program_Chairs · 2025-01-22

Reject